# Psychological Discomfort in Patients Undergoing Coronary Artery Bypass Graft (CABG) in West Bank: A Cohort Study

**DOI:** 10.3390/jcm13072027

**Published:** 2024-03-30

**Authors:** Osama Sawalha, Patrocinio Ariza-Vega, Fadwa Alhalaiqa, Sonia Pérez-Rodríguez, Dulce Romero-Ayuso

**Affiliations:** 1Department of Physical Therapy, Occupational Therapy Division, University of Granada, 18006 Granada, Spain; pariza@ugr.es (P.A.-V.); soniafis78@gmail.com (S.P.-R.); 2Instituto de Investigación Biosanitaria ibs. Granada, 18012 Granada, Spain; 3College of Nursing, Qatar University, Doha 2713, Qatar; f.alhalaiqa@qu.edu.qa; 4Brain, Mind and Behaviour Research Center (CIMCYC), University of Granada, Campus Universitario de Cartuja S.N., 18011 Granada, Spain

**Keywords:** depression, anxiety, stress, cardiovascular disease, coronary artery bypass graft surgery

## Abstract

**Background/Objetives**: Cardiovascular disease (CVD) remains a significant contributor to global morbidity and mortality rates. Coronary artery bypass graft (CABG) surgery is a critical intervention for patients with coronary artery disease, yet it poses psychological challenges that can impact recovery. **Methods**: This prospective cohort study, conducted across six hospitals in the West Bank/Palestine, aimed to assess changes in depression, anxiety, and stress levels among CABG patients and identify associated factors. The Arabic version of the Depression Anxiety Stress Scales (DASS-21) was administered before (one week) and after surgery (two and three weeks). **Results**: Of the 200 participants, 116 were men (58%). High levels of depression, anxiety, and stress were observed both before and after surgery, with statistically significant reductions in all these variables after surgery (*p* < 0.001). Regarding demographic factors, age displayed a weak positive correlation with depression (r = 0.283; *p* < 0.001), anxiety (r = 0.221; *p* = 0.002), and stress (r = 0.251; *p* < 0.001). Sex showed a weak correlation with stress pre-surgery (r = −0.160; *p* = 0.024). **Conclusions**: Patient outcomes could be improved by early identification and the provision of efficient treatments such as psychosocial therapy both before and after surgery.

## 1. Introduction

Cardiovascular disease (CVD) is one of the most important causes of morbidity and mortality in the world [1]. This has resulted in about 17.7 million fatalities worldwide each year and accounted for 31.5% of all deaths. Notably, most of these deaths (80%) occur in nations with middle- and low-income populations. In 2018, CVD accounted for 31.5% of deaths in Palestinian communities, making it the leading cause of death [2]. When considering the prognosis of patients with coronary artery disease (CAD), multivessel disease, and left main disease, coronary artery bypass graft (CABG) is still the best surgical intervention and treatment both in industrialized and developing nations [3,4,5]. Nevertheless, the risk of CVD and the expectations of having CABG are strongly correlated with psychological factors; stress at work and home, as well as low socioeconomic position, raise the risk of CVD and mental illnesses. Prior to surgery, patients often deal with stress brought on by fear of mortality, which activates the sympathetic nervous system, leading to increased blood pressure, heart rate, and energy consumption, ultimately hindering surgery and recovery. In addition, elevated levels of anxiety are associated with an intensification of pain sensations in patients after CABG, which may impede their recovery. Several studies have shown patients who were anxious pre-intervention with CABG reported experiencing more pain, less postoperative symptom relief, and more readmissions [6,7,8].

Health professionals are essential for improving patient outcomes and reducing psychological discomfort [9]. It has been shown that providing prospective patients undergoing CABG surgery with a 40 min orientation session can dramatically reduce their anxiety level before surgery. This emphasizes the significance of developing personalized care plans that are tailored to each patient’s needs. It is essential to give patients only the information they need and prevent information overload, as giving them too much information could make their situation worse [10,11].

Although our understanding of cardiac healthcare has advanced significantly, there are still a lot of gaps in the complete recommendations needed to properly educate patients before and after heart surgery [9]. In Palestine, there are no tertiary cardiovascular care programs offered by hospitals or community health nurses to provide and sustain post-discharge follow-up treatment for patients with coronary artery disease [12]. Consequently, after hospital discharge, patients make decisions independently without receiving official guidance from healthcare professionals.

From this perspective, it becomes imperative to investigate the psychological suffering that Palestinian patients having coronary artery bypass grafting (CABG) endure. This study aimed to describe the changes in depression, anxiety, and stress levels in patients who undergo CABG. The results of this study might lead to new approaches in the development and use of treatment approaches to help these patients and improve their outcomes after CABG in Palestine and other Arab nations.

## 2. Methodology

### 2.1. Design

This was a prospective cohort study conducted at selected hospitals in West Bank, Palestine, between February and November 2022.

### 2.2. Sample Size

The calculation of the sample size was based on a previous study, which an alpha error of 5% was assumed, a statistical power of 95%, and an effect size of 0.4 was estimated. In addition, 20% were considered as possible dropouts in the study, obtaining a total of 194 participants. G*Power 3.1 software was used to calculate the sample size [13].

### 2.3. Sampling and Setting

This study’s population comprised a convenience sample of cardiac patients who were candidates for CABG and were admitted to the six selected hospitals in Palestine. The inclusion criteria were (1) age between 50 and 85 years; (2) patients scheduled for CABG surgery; (3) a good level of oral and written comprehension to answer the questionnaires; (4) providing informed consent; (5) not taking antidepressant medications. The exclusion criteria were patients who underwent cardiac catheterization (CATH), stent surgery, cardiac arrhythmia, acute or past stroke, significant comorbidities affecting mental health (drug abuse, alcohol abuse), mild to severe cognitive impairment, neurological disorders (epilepsy, Alzheimer’s disease, dementia, Parkinson’s disease), end-stage renal disease and dialysis patients, peripheral vascular disease, or any other condition affecting their ability to answer questionnaires and interviews. These exclusions were made to avoid potential confounding factors.

### 2.4. Instruments

The present study used questionnaires and an interview as tools for data collection, with two sections: demographic characteristics, collecting information such as age, sex, marital status, number of children under 18, and educational level (primary, secondary, professional education/high school or university degree); and clinical data: previous illnesses, risk factors for coronary heart disease, blood pressure, smoking, alcohol consumption, an consumption of medications or other substances. The Depression, Anxiety, and Stress Scale-21 (DASS-21) Lovibond [14], was used to assess depression, anxiety, and stress.

### 2.5. The Depression, Anxiety, and Stress Scale-21 (DASS-21)

This is a self-reported questionnaire used to measure depression, anxiety, and stress. It includes 21 items. The original instrument was developed by Lovibond [14]. The instrument has three subscales, each one consisting of 7 items with 4 Likert scale options, measuring depression, anxiety, and stress symptoms over the last week. The scores for each subscale range from 0 to 21, with higher scores indicating higher symptom frequency and severity. For this study, the depression, anxiety, and stress subscales were used. The cut-off points for the depression subscale are 0–4 normal, 5–6 mild, 7–10 moderate, and ≥11 severe. The original instrument was found to be valid and reliable, with Cronbach’s alpha for the depression subscale being 0.91 [15]. The Arabic version has been validated; it is sensitive to youth, and it measures depression, anxiety, and stress in the same survey. The psychometric properties of the Arabic version were supported by different studies with different populations [16]. The above study supported the validity and reliability of the DASS using factor analysis, the universality of depression across cultures, and the ability to use the English norms for Arab populations. In that study, Cronbach’s alpha was 0.91 [17].

### 2.6. Procedure

Before surgery, physicians and nurses approached every patient scheduled for planned open-heart surgery and screened them using the inclusion and exclusion criteria. If the patient met the criteria, the study purpose was explained in detail, the participants’ rights were confirmed, and their questions were answered. The researcher invited all patients who met the criteria from the above six hospitals to participate in this study consecutively. Those who agreed to participate signed an informed consent and gave permission to review their medical records. After that, a meeting was conducted to collect the pre-surgery data. Two to three weeks after surgery, the researchers interviewed the patients by telephone. This approach made it possible to communicate with the patients directly and effectively, which sped up the process of gathering data.

### 2.7. Data Analysis

A level of statistical significance with a value of *p* < 0.05 (bilateral) was considered. Descriptive statistics were employed for the sociodemographic data of this study. To analyze pre–post differences in the depression, anxiety, and stress scales in each group, the paired *t*-test was used. The results are presented as the mean and standard deviation, along with the median with min/max values. To demonstrate the presence of correlation between these variables, the Pearson rank correlation test was utilized. All data analyses were conducted using IBM SPSS Statistics for Windows (version 23.0, IBM Corp., Armonk, NY, USA). A level of statistical significance with a value of *p* < 0.05 (bilateral) was considered.

### 2.8. Ethical Issues

This study was approved by the Research Ethics Committee from hospitals (IEC). All participants were informed about the purpose of this study and accepted their participation in it, giving their written informed consent. The data collected were treated according to the Declaration of Helsinki for research purposes, guaranteeing the protection and confidentiality of patient data. To preserve privacy, no identifiable data that could jeopardize the patient’s anonymity were provided.

## 3. Results

A total of 200 patients were included in the final sample. More than half of the sample were men (*n* = 116, 58%). The age ranged from 51 to 81 years, with a mean of 67 years (SD = 6.7). Most of the patients were married (*n* = 186, 93.3%), had between one and two children (*n* = 110, 55.3%), and had primary school education (*n* = 87, 43.5%). Regarding the type of surgery, all of them underwent scheduled surgery (*n* = 200, 100%). It is noteworthy that before the surgery, none of the 200 research participants died. However, it is noteworthy that six individuals passed after following therapy due to complications. A detailed description of sociodemographic and clinical variables is provided in Table 1.

The Table 2 shows significant changes in the main outcomes before and after CABG. There was a significant decrease in depression, anxiety, and stress levels (see Table 2).

### Correlation

Regarding demographic factors, before surgery, age displayed a positive correlation with depression (r = 0.283; *p* < 0.001), anxiety (r = 0.221; *p* = 0.002), and stress (r = 0.251; *p* < 0.001), where older individuals experienced slightly higher levels. Sex showed a weak correlation with stress pre-surgery (r = −0.160; *p* = 0.024). Men showed higher stress scores than women before surgery. Other demographic factors did not exhibit significant correlations with stress, anxiety, or depression. There were no statistically significant relationships after surgery.

The relationships between stress, anxiety, and depression in patients undergoing CABG are shown in Table 3. A moderate positive correlation was found between stress and anxiety before surgery, stress and depression before surgery, and depression and anxiety before surgery. In addition, a weak positive correlation was found between stress level pre- and post-surgery, stress level and anxiety post-surgery, and stress level and depression post-surgery. Similarly, a positive correlation was observed between depression pre-surgery and post-surgery.

## 4. Discussion

The present study reveals elevated levels of depression, anxiety, and perceived stress in patients before CABG and highlights the reduction in postoperative depression and stress in Palestinian patients. Moreover, changes in anxiety levels were observed before and after the intervention.

The results provided are consistent with the findings of other studies that have indicated a relationship between mental health and heart disease [9,18,19]. However, other studies have shown decreases in the levels of anxiety, depression, and stress before surgery [20,21]. According to these studies, Brazil and Ethiopia had reduced rates of anxiety prior to surgery because of societal and familial support. Furthermore, the discrepancy within our study can result from the instrument used to gauge the patient’s degree of anxiety prior to surgery [20,21].

### 4.1. Stress in Patients Undergoing CABG

The findings of our research support those of other studies that have found a connection between high stress levels both before and after CABG [20,22]. Before surgery, stress levels were found to be significantly higher than postoperative stress levels, indicating that the surgery may have contributed to stress reduction. More than 80% of patients who had CABG were reported to experience moderate to severe anxiety, according to research [23].

### 4.2. Depression in Patients Undergoing CABG

The severity of cardiac procedures and the effects of underlying heart problems are related to depression. The increased rates of depression might be conditioned by the dearth of specialized support systems in the hospitals, and it may be explained by the higher incidence of somatic symptoms in patients having [9].

Our findings found that depression levels before surgery were significantly higher than afterward, indicating that the surgery had a positive impact on reducing depressive symptoms during the recovery period (first three weeks) [3]. The results of our study are consistent with the results of previous studies where the level of depression was lower after surgery despite life-threatening factors and postoperative complications. However, the percentage of participants with moderate to severe depression in the current study was lower. One possible explanation for this difference is the timing of the depression measurement, which was performed three weeks after surgery in the current study, whereas in the earlier study, it was performed in the telemetry unit. Three weeks following surgery, when patients are released from the hospital and go home, they often see a quicker improvement in their health. It is possible that this change influenced their depression levels [24,25].

### 4.3. Anxiety in Patients Undergoing CABG

Similar to our results, previous studies have shown high levels of anxiety among patients before surgery [20]. Several factors influence anxiety in these patients. The most important factors appear to be the length of hospital stay in intensive care and concern about postoperative pain. There is also an increase in the rate of anxiety among patients who find it more difficult to follow the instructions of health professionals [20,22,23]. Other studies have shown a negative effect of coronary artery bypass graft (CABG) operation on the psychological status of patients, mostly due to concerns regarding pain and risks of mortality [26,27]. Psychological preparation before undergoing cardiac surgery seems to have positive effects on postoperative outcomes, at least under some conditions. Accordingly, patients who were highly anxious before the CABG procedure experienced more pain, lesser symptom relief following surgery, and frequent readmission [28]. Common fears that increase anxiety levels include fear of post-surgical pain, fear of deteriorating health, fear of myocardial infarction, and fear of CABG surgery [29].

In patients before undergoing CABG, anxiety is also a very important concern. Preoperative anxiety represents a significant concern. Prior research has suggested that the duration of the pre-surgical waiting period may contribute to heightened levels of anxiety. Additionally, studies have shown that patients’ functional and psychological states may deteriorate upon being identified as candidates for heart [30,31]. By incorporating the assessment of preoperative anxiety into the cardiac surgery preoperative procedure, nursing interventions can be developed to alleviate postoperative discomfort [32].

In our study, we found a significant association between age, stress, depression, and anxiety levels. According to Weiss et al., age is a major predictor of anxiety scores, with increasing age contributing to higher anxiety levels. Age is significantly related to anxiety level but not depression level. Parvan et al. reported that most patients over 60 should not have experienced high levels of anxiety because they are more likely to have experienced heart disease and hospitalization in the past, which has made them more accustomed to their current circumstances [33]. A negative relationship has also been found between age and anxiety levels before coronary artery bypass grafting (CABG) [27], with younger patients showing higher levels of anxiety, with a lack of health knowledge or with a lower educational level [34]. Patients with higher anxiety are more likely to be unable to follow the post-surgery plan and attention, resulting in adverse events [35,36].

All these findings underscore the psychological benefits of surgical intervention in addressing depression, anxiety, and stress. Providing patients with comprehensive healthcare, which includes managing potential consequences and addressing the psychological effects of treatment, is essential for achieving favorable outcomes [37]. The medical team must adopt various approaches to address psychological issues such as anxiety and depression, which may impact patients’ illnesses. To improve service quality, reduce anxiety and stress, and enhance overall patient quality of life, healthcare professionals can effectively provide psychological assistance and guidance to patients and their families [38]. Based on this study’s findings, it is imperative to prioritize providing patients with adequate psychological support both before and after the intervention. Before surgery, it may be beneficial to educate patients about the procedure and postoperative care, addressing any doubts and fears arising from a lack of information. Similarly, informing patients and their families about the appropriate post-surgery care plan appears to be relevant for optimizing recovery [39].

## 5. Limitations

Firstly, the sampling method used in this study was convenience sampling, which involved selecting patients undergoing CABG from various hospitals in Palestine. A second limitation is that these study participants were recruited only from specific hospitals in Palestine, thus limiting the generalizability of this study’s findings. Additionally, while the measurement tools used assessed the severity of depression, anxiety, and stress, they do not replace a formal clinical diagnosis. Therefore, the use of self-reported questionnaires may restrict the generalizability of this study’s results. Another significant limitation of our study is the lack of investigation into the potential relationship between preoperative stress levels and post-surgical complications, as well as recovery time among cardiac patients. Future studies should aim to determine whether preoperative stress levels are associated with postoperative complications and recovery duration among cardiac patients. Exploring this link in future research may provide valuable insights into the impact of psychological variables on surgical outcomes and recovery trajectories in cardiac patients.

## 6. Conclusions

There is a high prevalence of depression, anxiety, and stress both before and after CABG surgery. Early detection of depression, anxiety, and stress is essential to improve the recovery of these patients. It is important that medical personnel monitor stress, anxiety, and depression in these types of patients and, if necessary, implement preventive measures. The presence of psychosocial services in Palestinian hospitals would help in the rehabilitation of cardiac patients before and after surgery.

### Implications for Practice

This study’s findings have implications for the advancement of cardiac healthcare practices in the Palestinian context. It is important to know about psychological discomfort in cardiac patients. Its approach requires the provision of specialized training for health and rehabilitation personnel. Health professionals in cardiac surgery teams must have the necessary tools for the personalized psychological support that these patients require. Providing cardiac patients with knowledge about the repercussions of psychological stress on cardiac health could enable them to make lifestyle decisions, thus encouraging a proactive approach to health management. In essence, a comprehensive approach focused on the patient undergoing cardiac surgery could contribute to improving well-being and recovery after surgery.

## Figures and Tables

**Table 1 jcm-13-02027-t001:** Characteristics of the sample.

	N	%
**Sex**		
Male	116	58.0
Female	84	42.0
**Marital Status**		
Single/Divorced	14	7
Married	186	93.0
**Number of children**		
None	8	4.0
1	29	14.6
2	110	55.3
>3	52	28.1
**Education level**		
Primary	87	43.5
Secondary	60	30
Professional education/high school	22	11
University degree	31	15.5
**Hypertension**		
No	16	8.0
Yes	184	92.0
**Are you currently smoking?**		
Yes	144	72.0
No	56	28.0
**Are you alcoholic**		
No	200	100.0
**Do you take any medications**		
Yes	196	98.0
No	4	2.0
**Employment**		
Yes	22	11.0
No	178	89.0
**Are you worried for any economic problems?**		
Yes	195	97.5
No	4	2.0
**Surgery type**		
Scheduled	200	100.0

**Table 2 jcm-13-02027-t002:** Differences in preoperative and postoperative comparison of stress, depression, and anxiety.

	Pre-Surgery		Post-Surgery			
Variables	Mean	SD	Mean	SD	t	*p*
Stress	12.59	3.21	11.16	2.58	−6.114	<0.001
Depression	12.30	2.95	11.36	2.43	−4.855	<0.001
Anxiety	12.26	3.11	10.70	2.48	−7.176	<0.001

**Table 3 jcm-13-02027-t003:** Relationships between stress, anxiety, depression in CABG patients.

	Age	Sex	Post Stress	Pre Stress	Post Anxiety	Pre Anxiety	Post Depression
Post stress	0.037	0.022	1				
Pre stress	0.251 **	0.115	0.365 **	1			
Post anxiety	−0.076	−0.103	0.428 **	0.373 **	1		
Pre anxiety	0.221 **	0.127	0.269 **	0.689 **	0.423 **	1	
Post depression	0.125	−0.010	0.346 **	0.242 **	0.476 **	0.376 **	1
Pre depression	0.251 **	−0.160 *	0.394 **	0.639 **	0.432 **	0.686 **	0.229 **

** The correlation is significant at the 0.01 level (bilateral), * the correlation is significant at the 0.05 level (bilateral).

## Data Availability

The datasets are available from the corresponding author upon reasonable request.

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
