# Peer review of "Psychological Discomfort in Patients Undergoing Coronary Artery Bypass Graft (CABG) in West Bank: A Cohort Study"

_jcm, 2024, doi:10.3390/jcm13072027_

Round 1

Reviewer 1 Report

Comments and Suggestions for Authors

 Psychological discomfort in Patients Undergoing Coronary Artery Bypass Graft Surgery (CABG) in West Bank: A cohort Study

 Thank you for letting me review the current manuscript

Abstract: Please write the study conclusion clearly.

 Introduction: Line: 54-the end of the introduction; the authors did not  enrich their data with relevant studies, which weakens the section. Please enrich this section with relevant supported studies.

Sampling and setting: You mentioned that the study is a cohort, so possible confounding effects can be found. Post-CAPG the cause of increasing levels of anxiety, depression, and stress can be linked with other life situations. Here, you should write about how you stabilized the suspected confounding effect.

To include one governmental hospital and the remaining hospitals were private. This can confuse the study results as the psychological concerns are different. Please clarify.

Instruments, line 108, “An ad hoc questionnaire” , what the meaning of this phrase?

Page4 line 142: How the researchers complete the questionnaire? By phone?

Page4 line 161, writing error

Table 1: Please merge the variable: divorced, Number of children >4, to increase the statistical power, as these variables have few numbers.

Educational level variables: you write; Diploma and university level. These two variables have the same meaning. Use university level.

Did you come to work? What do you mean???

Add a correlation table.

Discussion: You write: The current study reveals high levels of depression, anxiety and perceived stress in 198 patients prior to CABG surgery and highlights the reduction of depression and postoperative stress involving Palestinian patients. However, I noticed that the most significant mean changes occurred in anxiety before and after the operation.

Page 7, line 255: Writing about tachycardia is not relevant.

Please review the sentences on page  7, lines 268-269.

Comments on the Quality of English Language

minor

Reviewer 2 Report

Comments and Suggestions for Authors

Dear authors,

I hope this message finds you well. I had the opportunity to read your paper with great interest. The study on changes in depression, anxiety, and stress levels among patients undergoing CABG is indeed intriguing, and I believe a few minor adjustments could enhance its clarity before publication.

In the Abstract section, it would be beneficial to specify the type and location of the six hospitals involved in the study. Additionally, clarification on whether the Arabic or English version of the scales was utilized would be helpful. Providing more details on how the sample size was collected could also improve the understanding of your methodology.

Regarding the Introduction section, I found the references to be updated and sufficient. However, adding a paragraph about the role of healthcare professionals in reducing mental health issues could enrich the context. Furthermore, replacing "Six" with "gender" would enhance clarity. It might be beneficial to define the main variables for better understanding.

In the Method section, additional details on data collection before and after the procedure would be beneficial. Mentioning that none of the 200 patients died and ensuring there were no missing patients would add completeness to your methodology. Additionally, clarification on why Spearman correlation was chosen for analysis, such as whether the data were normally distributed, would strengthen your approach. When presenting tables, it's essential to double-check the abbreviations for clarity and the significant values as note below the Tables. Lastly, the question "Did you come to work?" seems out of place; please consider editing or removing it for coherence.

Regarding the discussion section, did any of the patients included in this study had complications, thus anxiety and depression may increase. Right? Again, the role of HCPs can be beneficial for decreasing the depression and anxiety levels.

Overall, these suggestions aim to refine the paper and improve its clarity and comprehensiveness. I appreciate the opportunity to provide feedback and look forward to seeing the final version of your paper.

Comments on the Quality of English Language

Some editing is required. Coma, extra spaces and so on. 
